# A caveat to using wearable sensor data for COVID-19 detection: The role of behavioral change after receipt of test results

Jennifer L. Cleary[1,2]☯*, Yu Fang[1]☯, Srijan Sen[1,3], Zhenke Wu[4]

1 Michigan Neuroscience Institute, University of Michigan, Ann Arbor, MI, United States of America,
2 Department of Psychology, University of Michigan, Ann Arbor, MI, United States of America, 3 Department of Psychiatry, University of Michigan Medical School, Ann Arbor, MI, United States of America, 4 Department of Biostatistics, University of Michigan, Ann Arbor, MI, United States of America

☯ These authors contributed equally to this work.
* clearyjl@umich.edu

## Abstract

**Data Availability Statement:** The minimal underlying dataset is provided here as a Supporting Information file.

### Background

Recent studies indicate that wearable sensors can capture subtle within-person changes caused by SARS-CoV-2 infection and play a role in detecting COVID-19 infections. However, in addition to direct effects of infection, wearable sensor data may capture changes in behavior after the receipt of COVID test results. At present, it remains unclear to what extent the observed discriminative performance of the wearable sensor data is affected by behavioral changes upon receipt of the test results.

### Methods

We conducted a retrospective study of wearable sensor data in a sample of medical interns who had symptoms and received COVID-19 test results from March to December 2020, and calculated wearable sensor metrics incorporating changes in step, sleep, and resting heart rate for interns who tested positive (cases, n = 22) and negative (controls, n = 83) after symptom onset. All these interns had wearable sensor data available for > 50% of the days in pre- and post-symptom onset periods. We assessed discriminative accuracy of the metrics via area under the curve (AUC) and tested the impact of behavior changes after receiving test results by comparing AUCs of three models: all data, pre-test-result-only data, and post-test-result-only data.

### Results

Wearable sensor metrics differentiated between symptomatic COVID-19 positive and negative individuals with good accuracy (AUC = 0.75). However, the discriminative capacity of the model with pre-test-result-only data substantially decreased (AUC from 0.75 to 0.63; change = -0.12, p = 0.013). The model with post-test-result-only data did not produce similar reductions in discriminative capacity.

**Funding:** This study was supported by grants from the National Institute of Mental Health (R01MH101459) to SS and an investigator grant from Precision Health Initiative at University of Michigan, Ann Arbor to ZW and SS. JC was supported by T32HD007109 from the National Institute of Child Health and Development. The funders had no role in study design, data collection and analysis, decision to publish, or preparation of the manuscript. There was no additional external funding received for this study.

**Competing interests:** The authors have declared that no competing interests exist.

## Conclusions

Changes in wearable sensor data, especially physical activity and sleep, are robust indicators of COVID-19 infection, though they may be reflective of a person's behavior change after receiving a positive test result as opposed to a physiological signature of the virus. Thus, wearable sensor data could facilitate the monitoring of COVID-19 prevalence, but not yet replace SARS-CoV-2 testing.

## Introduction

Recent studies [1–8] suggest enormous public health potential of wearable sensors in capturing subtle within-person changes that indicate an infection, such as by SARS-CoV-2. Detection of infection via wearable data provides a potentially effective, scalable method of infection surveillance, through passive, non-invasive methods [9]. However, little is known whether the assessments of wearable sensors for SARS-CoV-2 infection to date is able to distinguish between two distinct streams of information—direct physiological effects of infection and behavioral changes secondary to learning confirmation of infection through receipt of test results [10]. Understanding the relative importance of these two streams of information in infection detection is critical to determining if infection surveillance is possible through wearable technology.

Specifically, common wearable metrics include an individual's resting heart rate, steps taken, and sleep duration [10]. While these measures are often impacted by illness directly via physiological changes due to infection, they may also be impacted by illness indirectly via the individual's coping behavior. Specifically, when a person prioritizes rest and sleep for recovery and symptom control, step count and sleep are particularly controllable by the individual. In the case of a respiratory pandemic like SARS-CoV-2, these behaviors may be further impacted by isolation or infection mitigation measures which encourage symptomatic individuals to stay in their residence. Notably, in the current pandemic, these mitigation measures are often recommended at symptom onset [11] but enforced upon a positive antigen or PCR result. Thus, symptomatic individuals with positive test results may exhibit differing behaviors due to isolation or quarantine than symptomatic individuals who test negative. This divergence would result in a specific wearable "signature" that distinguishes COVID-positive from COVID-negative individuals.

Studies that focus on the pre-symptomatic or asymptomatic period to detect changes in wearable metrics prior to testing likely escape this confound [7]; however, several others include data across the symptomatic and post-testing period. Whether this signature would be detectable without individuals receiving their test results is unknown and represents an important question for researchers and users of wearable devices in contextualizing wearables' optimal usage in infection detection.

Thus, we aim to assess whether the wearable metric differences between SARS-CoV-2 positive and negative symptomatic individuals are affected by COVID test result reception in addition to direct physiological change in a sample of medical interns, a population who is at high risk of COVID-19 exposure and incorporating date-of-results data to assess behavior change. Our data capitalize on a unique time in the pandemic where test results were neither immediate nor wholly unavailable. Thus, we are uniquely able to test within-individual changes in wearables data before and after receiving COVID-19 test results, contributing valuable information to assess the feasibility and utility of COVID-19 detection via wearable devices.

## Methods

### Study design and oversight

The Intern Health Study is a prospective cohort study that assesses mental health during the first year of residency training [12, 13]. Individuals starting residency in the 2019 and 2020 cohorts were invited to take part. Participating interns received a Fitbit Inspire HR or Charge 3 device (or $50 if they already have a Fitbit, (Fitbit Inc., San Francisco, CA); or an Apple Watch, (Apple Inc., Cupertino, CA)) and $60 in compensation. All participants provided written informed consent via a secure online survey, and the institutional review board at the University of Michigan approved the study.

### Study participants, recruitment, and enrollment

From April to December 2020, participants were sent multiple surveys that assessed whether they (1) exhibited any symptoms consistent with COVID-19 (e.g. fever, cough, shortness of breath, headache); (2) were tested for SARS-CoV-2 infection; (3) tested positive. Daily sleep duration, physical activity, and resting heart rate (RHR) were measured through Fitbit or Apple Watch throughout the first internship year. We focused on interns because this is a population that is likely to receive tests, receive test results quickly, and be more adherent to quarantine measures.

A total of 3,532 subjects participated in the 2019 and 2020 cohorts of Intern Health Study. Among them, 506 subjects experienced COVID-19-like symptoms between March 15 and December 2020 and of these, 379 reported being tested for SARS-CoV-2. There were 94 individuals who tested positive ("cases") and 285 individuals who tested negative ("controls"). We included in the analysis 22 cases and 83 controls who had step, sleep, and RHR data available for more than 50% of the days during baseline (21 to 7 days prior to symptom onset) and test (0–7 days after symptom onset) periods, respectively (Table 1). Participants were on average 28.5 +/- 2.81 years of age, and 50.5% (n = 53) of the sample were female.

## Methods

### Metrics definition

Participants were drawn from the 2019 and 2020 cohorts of the Intern Health Study. Study recruitment and procedures are detailed elsewhere [8]. Briefly, incoming first-year medical residents were surveyed throughout the pandemic from April to December 2020 and asked to report whether and when they experienced any potential COVID-19 symptoms, were tested, and their test results. The sample for this analysis included individuals who reported symptoms and a COVID-19 test, as well as at least 50% of the wearable data (collected through Fitbit or Apple Watch) during both baseline (21 to 7 days prior to symptom onset) and test (0 to 7 days after symptom onset) periods.

Following Quer et al. [6], we calculated metrics for sleep, activity, and resting heart rate (RHR), as well as an overall wearable sensor metric for each participant:

$$\textbf{RHRmetric} = \max(\text{dailyRHR[test]}) - \text{median}(\text{dailyRHR[baseline]})/\text{IQR}$$

$$\textbf{SLEEPmetric} = \text{mean}(\text{dailySLEEP[test]}) - \text{median}(\text{dailySLEEP[baseline]})/\text{IQR}$$

$$\textbf{STEPmetric} = \text{mean}(\text{dailySTEP[test]}) - \text{median}(\text{dailySTEP[baseline]})/\text{IQR}$$

$$\textbf{SENSORmetric} = \text{RHRmetric}/10 + \text{SLEEPmetric} - \text{STEPmetric}$$

**Table 1. Summary of key characteristics, metrics and COVID-19 test results among symptomatic participants.**

| | Including days after the test result (Scheme I) | | | Excluding days after the test result (Scheme II) | | | Excluding days before the test result (Scheme II) | | |
|---|---|---|---|---|---|---|---|---|---|
| | Test positive | Test negative | P value | Test positive | Test negative | P value | Test positive | Test negative | P value |
| **Demographics** | | | | | | | | | |
| Number of participants, N | 22 | 83 | - | 22 | 75 | - | 22 | 82 | - |
| Age, Mean (SD) | 28.1 (3.2) | 28.7 (2.7) | 0.49 | 28.1 (3.2) | 28.7 (2.7) | 0.46 | 28.1 (3.2) | 28.6 (2.7) | 0.54 |
| Female, N (%) | 7 (31.8%) | 46 (55.4%) | 0.06 | 7 (31.8%) | 41 (54.7%) | 0.09 | 7 (31.8%) | 45 (54.9%) | |
| Fitbit users, N (%) | 22 (100%) | 82 (98.8%) | - | 22 (100%) | 74 (98.7%) | - | 22 (100%) | 81 (98.8%) | - |
| AppleWatch users, N (%) | 0 (0%) | 1 (1.2%) | - | 0 (0%) | 1 (1.3%) | - | 0 (0%) | 1 (1.2%) | - |
| **Available days during baseline (IQR)** | | | | | | | | | |
| RHR | 13.4 (11.5–15) | 13.7 (13–15) | 0.55 | 13.4 (11.5–15) | 13.6 (13–15) | 0.66 | 13.4 (11.5–15) | 13.7(13–15) | 0.54 |
| Sleep | 14.3 (14–15) | 13.9 (13.5–15) | 0.30 | 14.3 (14–15) | 13.9 (13–15) | 0.24 | 14.3 (14–15) | 13.9 (13–15) | 0.30 |
| Activity | 14.9 (15–15) | 14.7 (15–15) | 0.16 | 14.9 (15–15) | 14.7 (15–15) | 0.14 | 14.9 (15–15) | 14.7 (15–15) | 0.16 |
| **Available days during symptomatic period (IQR)** | | | | | | | | | |
| RHR | 7.1 (7.0–8) | 7.3 (7–8) | 0.66 | 3.8 (2–6.75) | 3.0 (1–4) | 0.21 | 5.1 (3.25–7) | 5.2 (4–7) | 0.90 |
| Sleep | 7.6 (7.3–8) | 7.4 (7–8) | 0.12 | 3.9 (2–6.75) | 3.0 (1–4) | 0.15 | 5.5 (4–7) | 5.3 (4–7) | 0.57 |
| Activity | 7.9 (8–8) | 7.9 (8–8) | 0.89 | 4.0 (2–6.75) | 3.1 (1–4) | 0.21 | 5.7 (4–7) | 5.7 (4–7) | 0.93 |
| **Baseline mean (SD)** | | | | | | | | | |
| RHR (bpm) | 60.2 (5.6) | 65.3 (7.4) | <0.001 | 60.2 (5.6) | 65.6 (7.4) | <0.001 | 60.2 (5.6) | 65.3 (7.4) | <0.001 |
| Sleep (min) | 412 (38) | 411 (40) | 0.94 | 412 (38) | 411 (41) | 0.91 | 412 (38) | 412 (40) | 1 |
| Activity (steps) | 8650 (2808) | 8382 (2366) | 0.68 | 8650 (2808) | 8378 (2305) | 0.68 | 8650 (2808) | 8400 (2375) | 0.70 |
| **Symptomatic period mean (SD)** | | | | | | | | | |
| RHR (bpm) | 61.5 (5.6) | 65.7 (7.7) | 0.007 | 62.1 (5.5) | 66.2 (7.5) | 0.007 | 61.3 (5.7) | 65.6 (7.8) | 0.005 |
| Sleep (min) | 460 (67) | 428 (50) | 0.046 | 432 (83) | 411 (85) | 0.32 | 469 (75) | 433 (57) | 0.049 |
| Activity (steps) | 4948 (2223) | 7344 (2710) | <0.001 | 5756 (2411) | 7295 (3164) | 0.02 | 4638 (2494) | 7384 (2902) | <0.001 |
| **Mean change (SD)** | | | | | | | | | |
| RHR (bpm) | 1.3 (3.1) | 0.4 (2.3) | 0.18 | 1.9 (3.2) | 0.5 (2.5) | 0.07 | 1.1 (3.4) | 0.3 (2.4) | 0.30 |
| Sleep (min) | 47.9 (64.8) | 16.6 (48.2) | 0.044 | 20 (78) | 0.5 (91) | 0.33 | 56.9 (74) | 21.4 (49.2) | 0.043 |
| Activity (steps) | -3703 (3422) | -1038 (2333) | 0.002 | -2894 (3327) | -1083 (3140) | 0.03 | -4012 (3632) | -1016 (2599) | 0.001 |
| **Metric value (SD)** | | | | | | | | | |
| RHR | 0.36 (0.35) | 0.26 (0.27) | 0.20 | 0.26 (0.39) | 0.14 (0.25) | 0.18 | 0.33 (0.37) | 0.28 (0.27) | 0.24 |
| Sleep | 0.45 (0.60) | 0.19 (0.41) | 0.06 | 0.44 (0.95) | 0.24 (0.53) | 0.36 | 0.40 (0.62) | 0.17 (0.55) | 0.12 |
| Activity | -0.75 (0.69) | -0.18 (0.48) | 0.001 | -0.63 (0.79) | -0.17 (0.62) | 0.02 | -0.72 (0.68) | -0.18 (0.51) | 0.002 |
| Sensor Metric | 1.24 (1.1) | 0.40 (0.72) | 0.002 | 1.11 (1.43) | 0.42 (0.88) | 0.04 | 1.16 (1.03) | 0.37 (0.85) | 0.003 |

### Discriminative accuracy

We calculated ROC curves, AUC, sensitivity (SE), specificity (SP) for each metric to compare the intra-individual change in each metric with symptom onset between COVID-19 positive and COVID-19 negative individuals. To assess which part of the test period data is mainly responsible for the realized AUC, we calculated these parameters in three data schemes: Scheme I—using all the data in baseline and test periods; Scheme II—removing data on and after receipt of test results in test periods; Scheme III—removing data before receipt of test results in test periods.

### Conditional permutation tests

In order to test the statistical significance of the observed AUC decrease in Scheme II and III, we designed the one-sided conditional permutation tests in a way that breaks the link

between the indices of days removed during the test period and the dates of receiving the test results hence creating a null distribution that is adequate for assessing the statistical significance of the observed change in AUC. In particular, for each metric (RHR, sleep, activity, sensor) we performed the following steps in a computationally simple (milliseconds to run) framework:

**Step 1.** Calculate AUC based on all the baseline and test data;

**Step 2.** Remove part of the test data (on/after receiving the test results as in Fig 1B; OR before receiving the test results as in Fig 1C), and calculate a single AUC and the change from the AUC in Step 1;

**Step 3.** Create B = 1000 data sets, each by randomly removing the same amount of data for each person as in Step 2; based on each of B random reduced data sets, calculate an AUC and the difference from the AUC in Step 1, resulting in B = 1000 values of change in AUC;

**Step 4.** Compare the change of AUC in Step 2 against the null distribution of the change of AUCs in Step 3; Calculate the p-value by the observed fraction among the 1000 randomly reduced data sets that have AUC change less than or equal to the observed change in Step 2.

All analyses were conducted using R 4.0.2 (R Foundation for Statistical Computing).

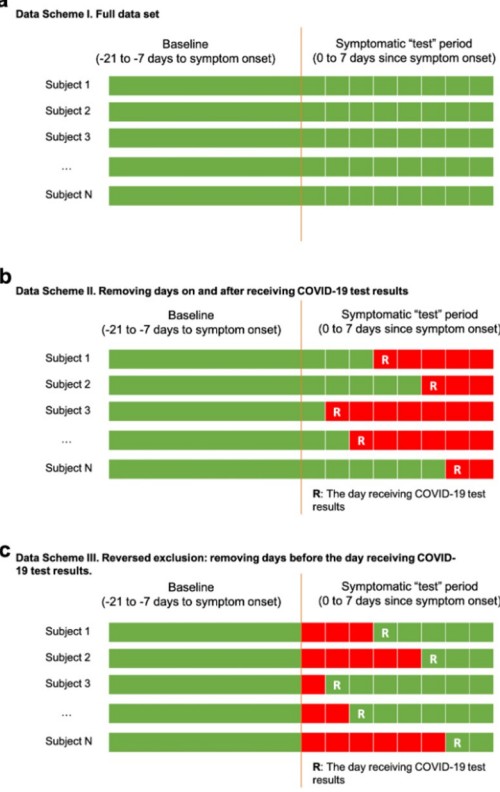

**Fig 1. Data schemes. a-c**, green: included data; red: excluded data; R: the day receiving test results: (**a**) include all data; (**b**) exclude data on and after the day receiving test results; (**c**) exclude data before the day receiving test results since symptom onset. Ninety-two subjects (87.6%) received their results within the symptomatic period (0 to 7 days after symptom onset).

## Results

Using all the data in baseline and test periods (Fig 1A), we observed that metrics of within-individual change discriminated cases from controls except for RHR (Fig 2A–2D). Sleep minutes increased more among cases than controls after symptom onset (mean change: 47.9 in cases, 16.6 in controls p = 0.044; area under the curve, AUC, based on SLEEPmetric = 0.66, 95% confidence interval, CI = 0.51–0.80). Cases reduced physical activity more than controls after symptom onset (mean change: -3,703 in cases, -1,038 in controls, p = 0.002; AUC based on STEPmetric = 0.75, 95% CI = 0.63–0.87). Mean change in RHR is higher in the cases (1.3 in cases, 0.4 in controls, p = 0.18) with the lowest discriminative ability based on RHRmetric (AUC = 0.63, 95% CI = 0.48–0.79). The combined metric based on all wearable sensor data results in an AUC of 0.75 (95% CI = 0.62–0.89).

To test whether the realized AUCs were mainly driven by the subset of data after receipt of test results, we conducted an analysis that removed data points on and after the result delivery date (Fig 1B). Compared with the previous analysis, we observed decreased discriminative ability (Fig 2E–2H) by SLEEPmetric (AUC = 0.60, 95% CI = 0.42–0.76), STEPmetric (AUC = 0.63, 95% CI = 0.49–0.78), and combined sensor metrics (AUC = 0.68, 95% CI = 0.50–0.82), but similar performance in RHR (AUC = 0.66, 95% CI = 0.51–0.86). The AUC based on STEPmetric experienced the largest decrease (delta = -0.12).

To assess whether the observed decrease in discriminative capacity is consistent with random data removal or systematic information loss, we further conducted one-sided conditional permutation tests for each metric (see S1 Dataset). In particular, the test assesses the null that, compared to random data removal, no additional decrease in AUC is caused by systematically removing data after receipt of test results. For the STEPmetric, the observed decrease in AUC (step 2, S1 Dataset) stands in the left tail of the reference distribution of change in AUC (step 3, Methods; observed change in AUC: -0.12, p = 0.013; Fig 3C), indicating the observed decrease in discriminative capacity upon removing post-result data is unlikely a chance event from data

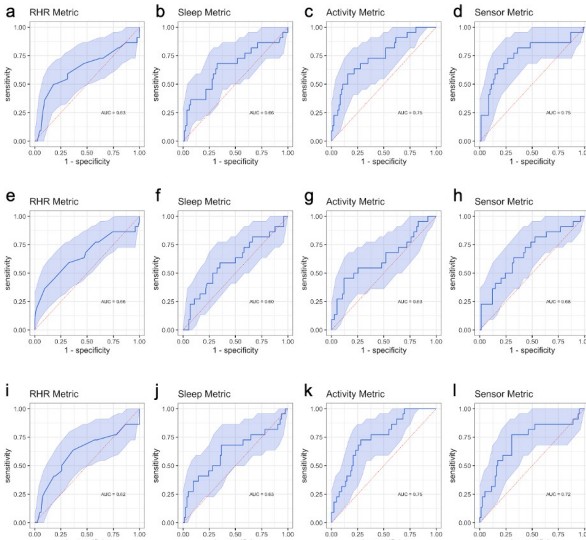

**Fig 2. AUCs based on RHR, sleep, activity and all-sensor metric derived from wearable sensors to differentiate symptomatic subjects who were tested positive and negative corresponding to data schemes I-III. (a-d)**: Scheme I —all data; (**e-h**) Scheme II—remove data on and after knowing the test results; (**i-l**): Scheme III—remove data since symptom onset and before the test results. For each data scheme in the row, the four panels are for RHR, sleep, activity and sensor metrics, respectively.

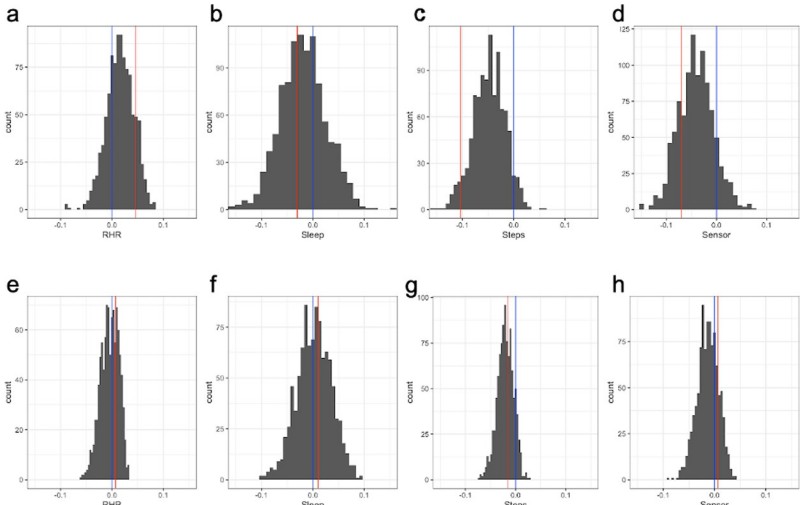

**Fig 3. One-sided conditional permutation test for assessing the null that, compared to random data reduction, no additional change in AUC caused by removing data.** (**a-d**) on or after receipt of test result and (**e-h**) in the symptomatic period and prior to receipt of test results. The random data removal and AUC calculation are done for RHR, sleep, step and the all sensor data, respectively (shown in four panels in each row). In each panel, the red line indicates the observed change of AUC; the blue line is at zero, indicating no change. The reference distributions are not centered at zero despite data removal being random; because on average there are more days after the receipt of test results than before, random data removal may still impact AUCs. For each metric, if a red line is at the left tail of the histogram, we conclude a statistically significant additional decrease in AUC.

reduction and hence the importance of post-result data. Although both cases and controls reduced average daily step counts after they became symptomatic, the reduction was significantly more among the cases after receipt of test results (mean change: cases -4,012, controls -1,016; p = 0.001) and more so than during the symptomatic period before receipt of test results (mean change: cases -2,894, controls -1,083, p = 0.03). For RHR and sleep metrics, we did not observe a statistically significant decrease in the AUC after removing the post-result data.

Finally, when only using the post-result-data in the test period (Fig 1C), the AUC is comparable to the all-data AUC for all metrics (Fig 2I–2L, RHRmetric: 0.62 vs 0.63; SLEEPmetric: 0.63 vs 0.66; STEPmetric: 0.75 vs 0.75; all-sensor: 0.72 vs 0.75), indicating no substantial loss of discriminative accuracy is incurred by only using post-result data when defining the metrics. We performed conditional permutation tests as above, but with the number of random days removed being the number of days prior to receipt of test results. No statistically significant decrease in AUCs was observed for any of the metrics (Fig 3E–3H).

## Discussion

We assessed the effect of test result receipt on wearable sensor data's discriminative capacity between symptomatic COVID-19 positive and symptomatic negative individuals using metrics of RHR, sleep, and steps created by previously published [6] methods. Our analysis reveals the discriminative accuracy of wearable data in COVID-19 detection can be explained by behavior changes after receiving test results. In particular, when removing data on and after receipt of test results, the discriminative capacity for the model based on the step metric drops significantly from the all-data model. A small though non-statistically significant drop was observed for the sleep metric, and no decrease was observed for the physiology-based RHR metric. This pattern indicates that the metric most susceptible to individuals' decisions and actions (steps) was most impacted by removing post-result data. This is consistent with behavior change after receiving COVID-19 test results.

Compared to symptomatic individuals who tested negative, symptomatic individuals who received a positive COVID-19 test may initiate stricter quarantine measures or more rest thus reducing physical activity. These measures may also lead to more sleep, but it appears that in the short term sleep is more resistant to change than physical activity during the test period, likely strongly regulated by circadian rhythms [14]. Comparatively, resting heart rate is the least modifiable by behavioral change upon receipt of test results [15].

This study has some limitations. First, our sample is a small subset of symptomatic subjects from a sizable cohort. In future studies, it is critical to aggregate data from multiple studies to further validate and study the variation in discriminative capacity with factors that may impact the propensity of behavioral change. Second, the cohort is likely not representative of the entire spectrum of population that may have access to both wearables and tests. However, the unique cohort of medical interns who are likely more adherent to quarantine measures strengthened the specific investigation addressed here. It is of interest to investigate the same question in a broader population. Third, the SARS-CoV-2 tests are not perfectly sensitive or specific. Knowledge about these test-related parameters will likely further improve the discriminative capacity. Fourth, recall of symptom onset date and test date might not be entirely accurate, but this population of medical interns is particularly primed to remember the dates due to workplace enforcements of symptom screening, testing, and compulsory quarantines.

Taken together, wearables data may facilitate the monitoring COVID-19 prevalence in conjunction with, but not yet replace, viral testing. The discrepancies in physical activity and sleep are robust indicators of COVID-19 infection and consistent with other reported results [10], though they may be reflective of a person's behavior change after receiving a positive test result as opposed to physiological changes due to the virus infection. To this end, changes in wearables data may not be fully able to serve as an early warning sign signaling individuals to seek COVID-19 testing; however, as the symptom onset versus positive antigen testing window changes with variants, this remains an important area for continued study. Additionally, as studies that incorporate other metrics such as self-reported symptoms [6] report additional discriminative capacity, objective but passively-collected measures of symptoms such as blood oxygen levels or body temperature may provide interesting new directions for detection algorithms.

In a future pandemic, passively-collected wearable data linked with test results may reveal distinct patterns of behavioral change across subpopulations. For example, lack of appropriate behavioral changes upon receiving test results may hurt discriminative accuracy based on wearable sensor data. Variation in the discriminative capacity of the step metric by age group may indicate differential levels of within-person change in activity. Groups with higher step-based discriminative capacity may have effectively quarantined after receiving their test results; while groups with lower step-based discriminative capacity may indicate either delay in their receiving the test results or difficulty and infeasibility in reducing physical activity. Subpopulations with lower observed discriminative capacity may benefit from more targeted public health policy innovations that may promote behavioral change, such as self-quarantine measures.

## Supporting information

**S1 Dataset.**
(CSV)

## Acknowledgments

We thank the interns and residency programs who took part in this study.

## Author Contributions

**Conceptualization:** Jennifer L. Cleary, Yu Fang, Srijan Sen, Zhenke Wu.

**Data curation:** Yu Fang.

**Formal analysis:** Jennifer L. Cleary, Yu Fang, Zhenke Wu.

**Funding acquisition:** Srijan Sen.

**Methodology:** Yu Fang, Zhenke Wu.

**Resources:** Srijan Sen, Zhenke Wu.

**Supervision:** Srijan Sen, Zhenke Wu.

**Validation:** Yu Fang.

**Visualization:** Jennifer L. Cleary.

**Writing – original draft:** Jennifer L. Cleary, Yu Fang, Zhenke Wu.

**Writing – review & editing:** Jennifer L. Cleary, Yu Fang, Srijan Sen, Zhenke Wu.

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
