## [Decision Letter · Decision Letter 0]

18 Feb 2022

PONE-D-21-40818A Caveat to Using Wearable Sensor Data for COVID-19 Detection: The Role of Behavioral Change after Receipt of Test ResultsPLOS ONE

Dear Dr. Cleary,

Thank you for submitting your manuscript to PLOS ONE. After careful consideration, we feel that it has merit but does not fully meet PLOS ONE’s publication criteria as it currently stands. Therefore, we invite you to submit a revised version of the manuscript that addresses the points raised during the review process. Note that failure to address properly the reviewer's concerns may still lead to rejection of the manuscript. 

We look forward to receiving your revised manuscript.

Kind regards,

Alessandro Borri

Academic Editor

PLOS ONE

Journal Requirements:

( This study was supported by grants from the National Institute of Mental Health (R01MH101459) to SS. JC was supported by T32HD007109 from the National Institute of Child Health and Development. The funders had no role in study design, data collection and analysis, decision to publish, or preparation of the manuscript.)

(This study was supported by grants from the National Institute of Mental Health (R01 MH101459) to S.S. and an investigator grant from Precision Health Initiative at University of Michigan, Ann Arbor to Z.W. and S.S.. J.C. was supported by T32HD007109.  We thank the interns and residency programs who took part in this study.)

( This study was supported by grants from the National Institute of Mental Health (R01MH101459) to SS. JC was supported by T32HD007109 from the National Institute of Child Health and Development. The funders had no role in study design, data collection and analysis, decision to publish, or preparation of the manuscript.)

(This study was supported by grants from the National Institute of Mental Health (R01 MH101459) to Dr. Srijan Sen and an investigator grant from Precision Health Initiative at University of Michigan, Ann Arbor to Drs. Zhenke Wu and Srijan Sen. The funders had no role in the design and conduct of the study; collection, management, analysis, and interpretation of the data; preparation, review, or approval of the manuscript; and decision to submit the manuscript for publication.)

( This study was supported by grants from the National Institute of Mental Health (R01MH101459) to SS. JC was supported by T32HD007109 from the National Institute of Child Health and Development. The funders had no role in study design, data collection and analysis, decision to publish, or preparation of the manuscript.)

Reviewers' comments:

Reviewer's Responses to Questions

**Comments to the Author**

1. Is the manuscript technically sound, and do the data support the conclusions?

Reviewer #1: Partly

Reviewer #2: Yes

2. Has the statistical analysis been performed appropriately and rigorously? 

Reviewer #1: I Don't Know

Reviewer #2: Yes

3. Have the authors made all data underlying the findings in their manuscript fully available?

Reviewer #1: No

Reviewer #2: Yes

4. Is the manuscript presented in an intelligible fashion and written in standard English?

Reviewer #1: Yes

Reviewer #2: Yes

5. Review Comments to the Author

Reviewer #1: The aim of this study was to utilised behavioral measures (sleep, steps, heart rate metrics) to identify whether individuals had tested positive for SARS-CoV-2, or had tested negative.

I commend the authors for collecting a large dataset - not an easy job.

The rationale for this study alone is not particularly impactful. I am not sure why utilising a wearable for this purpose would be useful, given the proliferation of self-testing kits. Given that the positive sample in this study are symptomatic, it is expected that their behavior will change.

I understand that the main utility for COVID-19 detection using wearable devices is to make the wearer aware of changes to their physiology, so that they can seek testing and reduce the spread. The analysis of this data do not allow for this, and would not provide insight into how to reduce the spread of the virus. It can be covered by general health recommendations of "get tested when you have symptoms".

My recommendations to the authors would be to use the amazing data they have collected to answer a more pertinent, more impactful question.

Reviewer #2: Add more keywords

Include a list of four to ten key words after the Abstract

The aim is not clear in the abstract section

Mention major contribution and paper organization clearly

More recent literature survey required, and compare with them

Improve the results and discussion section, add more sentences for proper justifying the works

Try to improve the usage of English grammar. The formatting, grammar and typo errors should be carefully checked before processing this article.

Measure accuracy of the model

Consider this paper Clinical and Laboratory approach to Diagnose COVID-19 using Machine Learning, Interdisciplinary Sciences: Computational Life Sciences, Intelligent Computing on Time-Series Data Analysis and Prediction of COVID-19 Pandemics, Pattern Recognition Letters

Measure computational complexity

Highlights major deliverables and future works

Why you consider wearable approach for this work

6. PLOS authors have the option to publish the peer review history of their article (what does this mean?). If published, this will include your full peer review and any attached files.

Reviewer #1: No

Reviewer #2: No

---

## [Author Response · Author response to Decision Letter 0]

4 May 2022

Reviewer 1: 

1. The rationale for this study alone is not particularly impactful. I am not sure why utilising a wearable for this purpose would be useful, given the proliferation of self-testing kits. Given that the positive sample in this study are symptomatic, it is expected that their behavior will change.

We thank the reviewer for bringing up the proliferation of self-testing kits and rapidly available test results, and agree that testing is likely the most reliable way to detect infection. We also now highlight that all individuals in our sample were symptomatic, while cases test positive for COVID and controls test negative (might have the flu or a cold instead). This allows us to compare across two groups of individuals both feeling subjectively unwell and therefore both reasonably likely to engage in self-care behaviors, but the COVID-positive individuals were more likely to be quarantined and take extra self-care. 

Location: Page 5, Lines 118-147; 152-56

2. I understand that the main utility for COVID-19 detection using wearable devices is to make the wearer aware of changes to their physiology, so that they can seek testing and reduce the spread. The analysis of this data do not allow for this, and would not provide insight into how to reduce the spread of the virus. It can be covered by general health recommendations of "get tested when you have symptoms".

This helpful comment from the reviewer highlighted the rationale for our study was not clear. We now have edited and reframed the introduction to emphasize that our primary goal is to urge caution in interpreting findings from the growing number of studies suggesting an important role for wearables in SARS-CoV-2 infection detection and alert researchers to a potential effect of test result receipt . We show that since no other studies provide information on test result receipt date, the current literature is unable to demonstrate conclusively that physiology change precedes test result receipt (which would be required in order to use wearables to alert users to seek testing). Rather, our results suggest that the signatures of COVID-positive wearable metrics are most pronounced AFTER the individual gets their test results, suggesting that an individual’s wearable metric change is dependent on the test result. 

Reviewer 2: 

1. Include a list of four to ten key words after the Abstract

Thank you for this guidance. We have included a list of keywords (wearable device, COVID-19 infection, COVID-19 detection, behavior change, model accuracy, public health, epidemiology) after the abstract.

Location: Abstract, Lines 102-103

2. The aim is not clear in the abstract section; mention major contribution and paper organization clearly

We are grateful to the reviewer for the feedback on clarifying the aims throughout the abstract and paper. We have updated the abstract, introduction, and discussion sections with a statement of our aims and main contributions. Specifically, our aim as well as the major contribution was being the first study to assess the role of COVID-19 test result receipt on wearable metrics’ discriminative capacity in distinguishing SARS-CoV-2 positive and negative symptomatic individuals. 

Location: Page 5, Lines 148-156

3. More recent literature survey required, and compare with them

We appreciate the reviewer’s direction and have updated the introduction with a more recent literature review. 

Location: Page 4-5, Lines 106-156

4. Improve the results and discussion section, add more sentences for proper justifying the works

We have updated the discussion section with contributions and justification for the study. 

Location: Page 16, Lines 336-347

5. Try to improve the usage of English grammar. The formatting, grammar and typo errors should be carefully checked before processing this article.

We appreciate the suggestion and have checked and corrected the use of English grammar, formatting and typos throughout the manuscript. 

6. Measure accuracy of the model

We agree the accuracy of the models are important and have used AUC and its 95% confidence interval as our primary measure of accuracy, noted in the Results text and in the Figures. 

7. Consider this paper Clinical and Laboratory approach to Diagnose COVID-19 using Machine Learning, Interdisciplinary Sciences: Computational Life Sciences, Intelligent Computing on Time-Series Data Analysis and Prediction of COVID-19 Pandemics, Pattern Recognition Letters

We appreciate the reviewer’s suggestions of additional papers to review and appreciated their background on machine learning methods; however, as our paper focuses primarily on behavioral change with test result date inclusion and less on developing a prediction algorithm or model, we restrained our literature review to those papers directly related to wearables metrics and change with COVID-19 infection. 

8. Measure computational complexity

We have included a sentence in Methods noting that these models run relatively quickly, on the order of milliseconds. 

Location: Page 12, Lines 239-240

9. Highlights major deliverables and future works

We have updated the aims and discussion to emphasize major findings and potential future directions.

Location: Page 5, Lines 148-152; Page 16, Lines 336-347; Page 17-18, Lines 383-396

10. Why you consider wearable approach for this work

Given the large interest and increasing body of work in using wearables for passive, real-time physiological data collection in infectious disease monitoring, we felt that investigating what factors affect changes in wearables data during an illness is a prudent and impactful contribution to the wearables literature in this pandemic and future pandemics. 

Editorial/Production:

Journal Requirements:

We have formatted the manuscript following the requirements in the style templates.

We have included information on participants’ written consent via an online survey and oversight via the University of Michigan IRB. 

( This study was supported by grants from the National Institute of Mental Health (R01MH101459) to SS. JC was supported by T32HD007109 from the National Institute of Child Health and Development. The funders had no role in study design, data collection and analysis, decision to publish, or preparation of the manuscript.)

We have updated our Funding Statement and include it in the cover letter. 

We have included the minimal underlying data set as a Supporting Information file in this revision.

(This study was supported by grants from the National Institute of Mental Health (R01 MH101459) to S.S. and an investigator grant from Precision Health Initiative at University of Michigan, Ann Arbor to Z.W. and S.S.. J.C. was supported by T32HD007109. We thank the interns and residency programs who took part in this study.)

( This study was supported by grants from the National Institute of Mental Health (R01MH101459) to SS. JC was supported by T32HD007109 from the National Institute of Child Health and Development. The funders had no role in study design, data collection and analysis, decision to publish, or preparation of the manuscript.)

Thank you for bringing this to our attention. We have removed the funding information from the Acknowledgements section and include the updated funding statement above.

---

## [Decision Letter · Decision Letter 1]

1 Aug 2022

PONE-D-21-40818R1A Caveat to Using Wearable Sensor Data for COVID-19 Detection: The Role of Behavioral Change after Receipt of Test ResultsPLOS ONE

Dear Dr. Cleary,

Thank you for submitting your manuscript to PLOS ONE. After careful consideration, we feel that it has merit but does not fully meet PLOS ONE’s publication criteria as it currently stands. Therefore, we invite you to submit a revised version of the manuscript that addresses all the remaining points raised during the review process.

We look forward to receiving your revised manuscript.

Kind regards,

Alessandro Borri

Academic Editor

PLOS ONE

Journal Requirements:

Reviewers' comments:

Reviewer's Responses to Questions

**Comments to the Author**

1. If the authors have adequately addressed your comments raised in a previous round of review and you feel that this manuscript is now acceptable for publication, you may indicate that here to bypass the “Comments to the Author” section, enter your conflict of interest statement in the “Confidential to Editor” section, and submit your "Accept" recommendation.

Reviewer #2: All comments have been addressed

Reviewer #3: (No Response)

2. Is the manuscript technically sound, and do the data support the conclusions?

Reviewer #2: Partly

Reviewer #3: Yes

3. Has the statistical analysis been performed appropriately and rigorously? 

Reviewer #2: Yes

Reviewer #3: Yes

4. Have the authors made all data underlying the findings in their manuscript fully available?

Reviewer #2: Yes

Reviewer #3: No

5. Is the manuscript presented in an intelligible fashion and written in standard English?

Reviewer #2: Yes

Reviewer #3: Yes

6. Review Comments to the Author

Reviewer #2: Improves the data analytics portion more robust way

Consider this below two papers to your introduction section and mention it Clinical and Laboratory approach to Diagnose COVID-19 using Machine Learning, Interdisciplinary Sciences: Computational Life Sciences,

Intelligent Computing on Time-Series Data Analysis and Prediction of COVID-19 Pandemics, Pattern Recognition Letters

Add future scope

Justify the relevancy of the work

Reviewer #3: The manuscript has largely improved compared to the previous version. The reviewers' suggestions were accepted and implemented in an appropriate way by the authors. The overall result was an increase in the quality of reporting. However, I think it is necessary to argue in more detail how the results of the article are consistent with the rest of the literature (line 313 of the revised manuscript), in order to discuss in depth the results of the original article in the light of the existing literature.

7. PLOS authors have the option to publish the peer review history of their article (what does this mean?). If published, this will include your full peer review and any attached files.

Reviewer #2: No

Reviewer #3: No

---

## [Author Response · Author response to Decision Letter 1]

23 Sep 2022

Reviewer 2: 

1. Consider this paper Clinical and Laboratory approach to Diagnose COVID-19 using Machine Learning, Interdisciplinary Sciences: Computational Life Sciences, Intelligent Computing on Time-Series Data Analysis and Prediction of COVID-19 Pandemics, Pattern Recognition Letters. Add future scope.

We appreciate the reviewer’s suggestions of additional papers to review and appreciated their background on machine learning methods; these papers do focus on aspects of machine learning applications in detecting COVID-19 but in blood samples and social networking sites, respectively. Our analysis focuses on the changes in resting heart rate, physical activity, and sleep as measured by smartwatch/wearable technology during COVID-19 infection before and after receiving PCR test results. As our paper focuses primarily on behavioral change with test result date inclusion and less on developing a prediction algorithm or model, we restrained our literature review to those papers directly related to wearables metrics and change with COVID-19 infection. Additionally, we have incorporated text to better situate our paper in the existing literature and suggest future directions in Line 292 of the Discussion (see below). 

Reviewer 3: 

1. The manuscript has largely improved compared to the previous version. The reviewers' suggestions were accepted and implemented in an appropriate way by the authors. The overall result was an increase in the quality of reporting. However, I think it is necessary to argue in more detail how the results of the article are consistent with the rest of the literature (line 313 of the revised manuscript), in order to discuss in depth the results of the original article in the light of the existing literature.

Thank you for your comments on our revisions and direction to expand on our results in context. We have added a discussion of where our paper fits within a review of the wearables and infection detection literature and papers conducting similar analyses to ours. 

Line 292-299: “To this end, changes in wearables data may not be fully able to serve as an early warning sign signaling individuals to seek COVID-19 testing; however, as the symptom onset versus positive antigen testing window changes with variants, this remains an important area for continued study. Additionally, as studies that incorporate other metrics such as self-reported symptoms(6,9) report additional discriminative capacity, objective but passively-collected measures of symptoms such as blood oxygen levels or body temperature may provide interesting new directions for detection algorithms.”

---

## [Decision Letter · Decision Letter 2]

26 Oct 2022

A Caveat to Using Wearable Sensor Data for COVID-19 Detection: The Role of Behavioral Change after Receipt of Test Results

PONE-D-21-40818R2

Dear Dr. Cleary,

We’re pleased to inform you that your manuscript has been judged scientifically suitable for publication and will be formally accepted for publication once it meets all outstanding technical requirements.

Kind regards,

Alessandro Borri

Academic Editor

PLOS ONE

Additional Editor Comments (optional):

Reviewers' comments:

Reviewer's Responses to Questions

**Comments to the Author**

1. If the authors have adequately addressed your comments raised in a previous round of review and you feel that this manuscript is now acceptable for publication, you may indicate that here to bypass the “Comments to the Author” section, enter your conflict of interest statement in the “Confidential to Editor” section, and submit your "Accept" recommendation.

Reviewer #2: All comments have been addressed

Reviewer #3: All comments have been addressed

2. Is the manuscript technically sound, and do the data support the conclusions?

Reviewer #2: Yes

Reviewer #3: (No Response)

3. Has the statistical analysis been performed appropriately and rigorously? 

Reviewer #2: N/A

Reviewer #3: (No Response)

4. Have the authors made all data underlying the findings in their manuscript fully available?

Reviewer #2: Yes

Reviewer #3: (No Response)

5. Is the manuscript presented in an intelligible fashion and written in standard English?

Reviewer #2: Yes

Reviewer #3: (No Response)

6. Review Comments to the Author

Reviewer #2: A serious proofreading of the manuscript is required

Follow the journal guidelines and plagiarism policy strictly

Reviewer #3: In my opinion, this original article is ready for publication because the authors have adequately addressed my comment.

7. PLOS authors have the option to publish the peer review history of their article (what does this mean?). If published, this will include your full peer review and any attached files.

Reviewer #2: No

Reviewer #3: No

---

## [Editor Report · Acceptance letter]

19 Dec 2022

PONE-D-21-40818R2 

A Caveat to Using Wearable Sensor Data for COVID-19 Detection: The Role of Behavioral Change after Receipt of Test Results 

Dear Dr. Cleary:

I'm pleased to inform you that your manuscript has been deemed suitable for publication in PLOS ONE. Congratulations! Your manuscript is now with our production department. 

Kind regards, 

on behalf of

Dr. Alessandro Borri 

Academic Editor

PLOS ONE